# Accelerating Density Functional Calculation of Adatom Adsorption on Graphene via Machine Learning

**DOI:** 10.3390/ma16072633

**Published:** 2023-03-26

**Authors:** Nan Qu, Mo Chen, Mingqing Liao, Yuan Cheng, Zhonghong Lai, Fei Zhou, Jingchuan Zhu, Yong Liu, Lin Zhang

**Affiliations:** 1School of Materials Science and Engineering, Harbin Institute of Technology, Harbin 150001, China; 2National Key Laboratory of Science and Technology on Advanced Composites in Special Environments, Harbin Institute of Technology, Harbin 150001, China; 3Center of Analysis, Measurement and Computing, Harbin Institute of Technology, Harbin 150001, China; 4Biological Physics, Department of Physics and Astronomy, University of Manchester, Oxford Road, Manchester M13 9PL, UK

**Keywords:** graphene, adsorption, machine learning, DFT calculation

## Abstract

Graphene has attracted significant interest due to its unique properties. Herein, we built an adsorption structure selection workflow based on a density functional theory (DFT) calculation and machine learning to provide a guide for the interfacial properties of graphene. There are two main parts in our workflow. One main part is a DFT calculation routine to generate a dataset automatically. This part includes adatom random selection, modeling adsorption structures automatically, and a calculation of adsorption properties. It provides the dataset for the second main part in our workflow, which is a machine learning model. The inputs are atomic characteristics selected by feature engineering, and the network features are optimized by a genetic algorithm. The mean percentage error of our model was below 35%. Our routine is a general DFT calculation accelerating routine, which could be applied to many other problems. An attempt on graphene/magnesium composites design was carried out. Our predicting results match well with the interfacial properties calculated by DFT. This indicated that our routine presents an option for quick-design graphene-reinforced metal matrix composites.

## 1. Introduction

Graphene, as a single layer two-dimensional material, has attracted increased interest in catalysis, hydrogen storage, and many other fields [1,2,3,4]. For heterogeneous single-atom catalysts, graphene has been reported to be a superb support. Graphene has also been considered as a nano-reinforcement for composites [5,6,7]. The interfacial properties of graphene are significant [8], and considerable research on the adsorption process [9,10,11,12] has been carried out. The hitherto-investigated elements used in metal single-atom supported by graphene include over half of the metal elements (37/64) in the first to sixth periods [2]. Due to the huge number of combinations of adatoms and adsorption sites, which include the top (T) site, the bridge (B) site and the hollow (H) site, computational methods are widely used in accelerating possible combination selection in order to save precious time and resources. 

The most common of these many simulation methods is a first-principle calculation based on density functional theory (DFT) [13,14,15,16], because of its advantages in assessing the chemical and mechanical stability of novel materials. However, the computing cost of a DFT calculation is huge. In order to balance the need for calculation accuracy and the computing cost, we employed the machine learning (ML) method to accelerate the DFT calculation.

ML is a widely used data-driven statistics approach that balances the computing cost and calculation accuracy well in the development of novel materials. In order to enhance the possibility of establishing accurate predictive rules, many attempts have been performed with small datasets, which are typically used in materials research [17,18,19]. The advances in applying ML to scientific research provide new opportunities for balancing calculating accuracy and costs [20,21,22]. This is because the ML workflow bypasses the computationally costly step of solving the Schrödinger equation [23]. 

A typical workflow for accelerating a DFT calculation with ML consists of three necessary steps. First, the atomic structure must be transformed into a numerical descriptor. The descriptor usually contains both atomic and location information for each atom. This descriptor is then used as an input for an ML model. After training the ML model, the properties prediction is provided as the final step of the workflow. Compared with solving the Schrödinger equation, the computational speed of the ML process is faster, by several orders of magnitude.

In this work, we present an ML workflow to accelerate the DFT calculation of adatom adsorption on graphene. This workflow could be applied generally to other DFT calculations of adsorption problems. With ML speed, our workflow could provide an adsorption properties prediction with accuracy close to the DFT calculation. The workflow consists of two main parts. One main part is a DFT calculation routine for generating a dataset automatically. This involves three main processes: adatom random selection, modeling adsorption structures automatically, and DFT calculation. As an example, we settled a single-atom adsorption on pristine graphene. A small dataset was generated with 34 adatom-graphene adsorption cases. Our workflow realized a quick structure search in DFT calculation accuracy, using just one-third of the DFT calculation cases with adatom elements from the entire periodic table.

## 2. Modeling and Dataset

### 2.1. First Principles Calculations

Our calculation routine can be used in different adsorption types, such as adsorption on pristine, defective, or decorated graphene and on yop (T), bridge (B), hollow (H), or edge sites, by changing the initial structures. Here, we take the adatom adsorption on pristine graphene as an example. The adatom adsorption structure is shown in Figure 1a. The adatom was settled on the highly symmetric graphene H site [4]. The DFT calculation routine, which includes random adatom selection (except for the third transition metal elements), automatic atom structure modeling, and DFT calculation, covering 34 adatom adsorption cases. The first-principles calculations were performed with DFT using the Cambridge Sequential Total Energy Package (CASTEP). The generalized gradient approximation (GGA) in the revised Perdew–Burke–Ernzerhof (RPBE) format and the projector-augmented wave (PAW) method were employed in all calculations. A plane wave basis with a cut-off energy of 525 eV and 3×3×1 k-sampling in the Brillouin zone were used for all calculations. Ultrasoft pseudopotentials were employed for a description of the interaction between the ionic cores and the valence electrons. Energy and structure optimizations were carried out on a 3×3 supercell, and 15 Å of a vacuum layer in a perpendicular direction were included in supercells to avoid non-physical interactions between neighboring unit cells. The adsorption energy, $E_{ads}$, is written as
(1)$$E_{ads}=E_{A/G}-(E_{A}+E_{G})$$
where $E_{A/G}$ is the total energy of adatom adsorbed on graphene layer, $E_{A}$ is the energy of an isolated adatom, and $E_{G}$ is the energy of the pristine graphene layer.

Figure 1b,c encompasses the adsorption energies and distances according to the DFT calculation (the details are provided in Appendix A). The red blocks represent a relatively strong interaction, as shown in Figure 1b, while in Figure 1c, the red blocks represent a far adsorption distance. Our DFT-calculation result showed a positive correlation. The adsorption energies for transition metals of groups VI, VII, XI, and XII were rather low, due to the semi-occupancy or full occupancy of d orbitals. This caused evolution along the third series to appear as a camel-hump shape, as shown in Appendix A. It should be noted that the absolute values of adsorption energies of elements with high accuracy in gaining and losing electrons—for example N, F, and Rb—are large, while the adsorption distances are also large, due to the different adsorption mechanisms with adatoms of elements in the middle of the periodic table. Taking Cr and F as examples, the electron density difference plots are shown in Appendix A. Twenty-two adatoms were selected randomly and uniformly in the DFT calculation routine as the training dataset; 5 observations were selected as the validation dataset; and the other 7 observations were selected as the test dataset.

### 2.2. Feature Engineering

The atomic characteristics of adatoms were taken into consideration as input features. The inputs were reduced by feature engineering to simplify our model. A Pearson correlation coefficient matrix was employed to perform the correlation analyses. The Pearson correlation coefficient $\rho_{X,Y}$ between two feature datasets $\left\{X_{i}\right\}$ and $\left\{Y_{i}\right\}$ was calculated as
(2)$$\rho_{X,Y}=\frac{\sum (X_{i}-\bar{X})(Y_{i}-\bar{Y})}{\sqrt{\sum (X_{i}-\bar{X}{)}^{2}\sum (Y_{i}-\bar{Y}{)}^{2}}}$$
where $\bar{X}$ and $\bar{Y}$ represent the average values of each feature over the respective dataset. Figure 2 is the heat map of the matrix among both the input and output features, in which the red blocks and the blue blocks represent positive and negative correlations, respectively. Appendix A lists 16 input features that were considered, including group, period, atomic number, relative atomic mass and several atomic features, including atomic radius, electronegativity, ionization energy, electron configuration, and electron affinity of adatoms. The atomic properties were extracted from the periodic table database of the Royal Society of Chemistry (https://www.rsc.org/periodic-table/ (accessed on 1 March 2023)). The pair correlation of inputs and adsorption properties is shown in Figure 2. The input elemental properties, which have relatively stronger correlation (>0.22) with adsorption properties, including atomic number, first ionization energy, element position (including group and period), electron affinity, and electron configuration, were selected as inputs for our model. Electron affinity had the strongest correlation with adsorption distance. However, some elements, such as Be, N, Mg, did not have stable electron affinity. Fortunately, Pauling electronegativity has a strong correlation with electron affinity (0.8444). Therefore, Pauling electronegativity was selected as one of our inputs. Atomic number and electron configuration were represented by element position due to their close correlation. Due to the missing data for second ionization energies, third ionization energies, and electron affinity, the following feature selection considered only the other 13 input features.

It is often useful to choose a number of components to minimize the prediction error, especially for small dataset problems. Here, principal components regression (PCR) was employed to select the number of components. Figure 3a shows the estimated mean squared prediction error (MSPE) curves by cross-validation, in this case using 10-fold cross-validation. The MSPE curves show sharp increases when the number of components exceeds 5. The MSPE curves prove that simply using a large number of components will do a good job in fitting the current observed data, but such a strategy leads to overfitting. Fitting the current data too well results in a model that does not generalize well to other data, and provides an overly optimistic estimate of the prediction error. According to feature selection, the model with 4 or 5 inputs shows the low MSPE of both the adsorption energy output and the adsorption distance output. We scored these 13 inputs in our model using F-tests, as shown in Figure 3b. The feature importance bars were ranked by the scores of adsorption distance output. Although atomic number, relative atomic mass, and s-orbital electron configuration gained high importance scores, they showed great correlation with the element site (>0.9), according to the correlation analysis. Therefore, these three features were not selected. Finally, site code (including group and period), first ionization energy, Pauling electronegativity, and a van der Waals radius were selected as inputs for our model.

### 2.3. Machine Learner Design

Our machine learner is schematically presented in Figure 4. We used artificial neural networks (ANN) to build the prediction engine. As a common ANN procedure [24,25,26,27], the machine learner consists of three parts: an input dataset, a targeted dataset, and the ANN algorithm, as shown in the blue box in Figure 4. The locations of the elements, the relative atomic mass, and several atomic features, including the atomic radius, electronegativity, ionization energy, and the electron affinity of adatoms, were taken into consideration for the input dataset. Our dataset consisted of 34 observations, which is a typically small dataset in ML. The small dataset presents quite a high risk of overfitting or underfitting [28,29]. A less complex ML model is more friendly to a small dataset. Therefore, feature engineering was carried out in order to remove redundant features. The targeted dataset was the adsorption performances obtained from the DFT calculation routine, including adsorption energy and adsorption distance. Back propagation neural networks (BPNNs) were employed to establish the prediction engine, in which the parameters are optimized by a genetic algorithm (GA). Single hidden layer BPNNs with 3–17 neurons were selected. In addition, using weights and thresholds optimized by the GA improved the fitting performances in BPNNs with small datasets. The parameters of the GA are shown in Appendix A. This machine learner enables the adatoms adsorption performances predictions of all the adatoms on the periodic table (except noble gases, lanthanides, and actinides).

## 3. Results and Discussion

Neurons in the hidden layer were selected by a parameter test in which the sum of squared error (SSE) was employed to judge the predictability of the model. Figure 5a shows the SSE variation of the whole dataset with 3–17 neurons in the hidden layer. The SSE variation indicated that the high complexity of the model presented the problem of overfitting and finally caused a decrease in accuracy, especially for the small dataset. According to the parameter test (Figure 5a), the 5×8×2 network was selected, due to the fact that the SSE curves of both two outputs showed minimums. The fitness curve of our selected BPNN optimized by GA is plotted in Figure 5b. Here, fitness was defined as the sum of absolute error of outputs after normalization. When evolutional generation progressed, fitness declined. After 30 generations, the pace of fitness decline slowed, which meant that our BPNN had found its suitable weights and thresholds. This could also be investigated in the regression scatter of the entire dataset, as shown in Figure 5c. The regression scatters with the train, validation, and test datasets are plotted in Appendix A. The data were normalized to −1 to 1. The *x* axis titled target refers to the adsorption properties calculated by DFT, and the *y* axis titled output refers to the properties calculated by our ML model. The data were distributed evenly around the fitting curve. The vertical intercept was close to 0, and the slope of the regression equation was close to 1. In addition, the correlation coefficient was 0.77. These three parameters proved that there was only a small difference between the adsorption properties calculated by our ML model and those calculated by DFT, which meant that our model optimized by the GA provided good predictability. The absolute percentage error distribution, as shown in Figure 5d, also proved that our model provided an accurate prediction via DFT calculation of adatom adsorption properties. The predicting results, with a percentage error below 20%, were 52.94% (36/68). In addition, predicting errors of more than 79% (54/68) of the data were below 40%. In our selected model, the training MAPE was 30.0% and the testing MAPE was 38.1%. Figure 5e provides a comparison between the adsorption properties of the testing dataset calculated by ML and DFT, in which we can see clearly that the ML calculation results matched well with the DFT calculation results. The difference between the DFT and ML calculations of F and N adatoms may be caused by the difference between the types of adsorption. For metal adatoms, there are trends to form physical adsorption. However, there are trends to form chemical adsorption for non-metal adatoms. These differences between DFT and ML, which are caused by atomic special characteristics, are within the accuracy of DFT calculation.

A further validation of our model’s predictability of extrapolation of the whole periodic table was obtained, as shown in Figure 6. The camel-hump shape curves of transition metal in the fourth, fifth, and sixth periods reappear well in our ML adsorption energy predicting, as presented in Appendix A. The unique character of elements with a powerful ability to gain or lose electrons also presented well. The ML predicting results of adatoms of elements in the southeast and northwest part of periodic table showed both a large adsorption distance and a large absolute value of adsorption energy, which presented the same characteristics as the DFT calculating results. According to the prediction results, new structures could be selected for their adsorption properties. For example, N, B, and Al predict the same adsorption properties as C. Therefore, these elements could be selected to remediate graphene. Interactions between noble metal elements, such as Au, Ag, and Pt, with graphene are weak. Compared with Y, Zr, and Hf, noble metal adatoms become gathered when they are adsorbed on graphene. These predictions could be confirmed by literature [2,30,31,32].

We examined our predicting results in graphene/magnesium composites design. Four alloying elements (Al, Zn, Ca, and Li) that are popular in magnesium alloys were selected. We studied the interfacial interactions of pristine graphene/Mg + M (M = Al, Zn, Ca, Li) using the CASTEP code within the framework of DFT. The calculation details are provided in the Appendix A. The interfacial distance ($d_{inter}$) after full geometry relaxation, cohesive energy ($E_{coh}$) of graphene/Mg + M interfaces, and adsorption energy of the correlated atom on graphene predicted by our network are listed in Table 1. The rank of cohesive energies of the four graphene/Mg + M interfaces, which represents the stability of the interface, matched well with the predicting results of adsorption energy by our ML network. The mismatch between interfacial distance and adsorption distance was due to interactions between Mg and M. The electron density difference and the partial density of states (the details of which are shown in the Appendix A) were also calculated. Our DFT-calculation results indicated that the graphene/Mg + Al interface showed better interfacial stability, which meant that introducing elements whose adsorption energy is larger than Mg could enhance the interfaces between graphene and Mg alloys.

We recorded the DFT-calculation computing costs in Appendix A. The average cost was 352.47 core-hours (cost hours running in parallel on 1 core) per adatom–graphene adsorption case. The ML model training cost was 359.393 core per second and the predicting cost was 12.922 core per second. Compared with the DFT-calculation computing cost, the cost of ML modeling and predicting was a couple of orders of magnitude less, which was negligible. The total computing cost of our workflow was 24 times 352.47 core-hours plus the ML modeling cost, which was only one-third of the DFT calculation cost of all the adsorption cases covering the entire periodic table. This indicates that our workflow could accelerate the DFT calculation of adatom–graphene adsorption effectively, with about two-thirds savings in calculating costs.

## 4. Conclusions

We proposed an adsorption structure selection workflow based on first-principle calculation and ML. This included a DFT calculation routine to generate a dataset automatically and a BPNN to speed up the calculation in DFT accuracy. By using our workflow, a DFT accuracy adsorption-properties prediction with a sharp lifted speed covering the whole periodic table was realized. 

Our workflow provides a general computing routine to accelerate DFT calculations. By changing the initial adsorption structure model, our workflow could be used in many other adsorption problems. Here, the single-atom adsorption on pristine graphene was employed as an example. The DFT calculation routine generated a small dataset for an adsorption problem, which included adatom random selection, modeling adsorption structures automatically, and DFT calculation. We calculated the adsorption properties, including adsorption energy and adsorption distance, of 34 adatom-graphene adsorption cases using our DFT calculation routine. Twenty-five cases were selected as the dataset for training the BPNN. Input feature selection and network feature optimizing were carried out to lower the risks of overfitting and underfitting, which are typical problems in ML with a small dataset. Finally, we built a network with 33.38% MAPE. By using our network, a quick fully covered periodic table prediction of adatom-graphene adsorption properties was provided with DFT calculation accuracy. 

The adsorption properties prediction using an ML network accurately reflects the variation caused by atomic characteristics. Our workflow provides a quick structure search and properties prediction of adsorption problems in the whole periodic table by using only one-third of the element cases, which could save about two-thirds of computing costs, compared with the cost of DFT calculation. We applied our ML network to a graphene Mg composites design case, which was well matched with DFT results. This indicated that our network has good predictability and our pipeline could help accelerate composites design.

## Figures and Tables

**Figure 1 materials-16-02633-f001:**
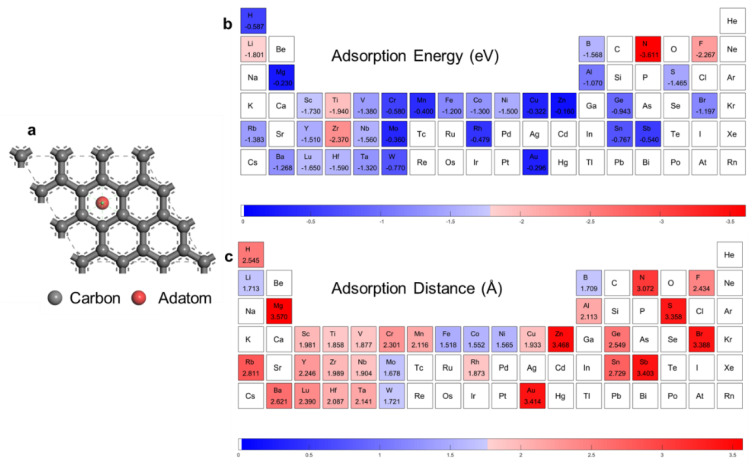
DFT calculation results. (**a**) Adatom adsorption structures. The adatom is put at the hollow site. Data visualization of first principle calculated. (**b**) Adsorption energy. (**c**) Adsorption distance.

**Figure 2 materials-16-02633-f002:**
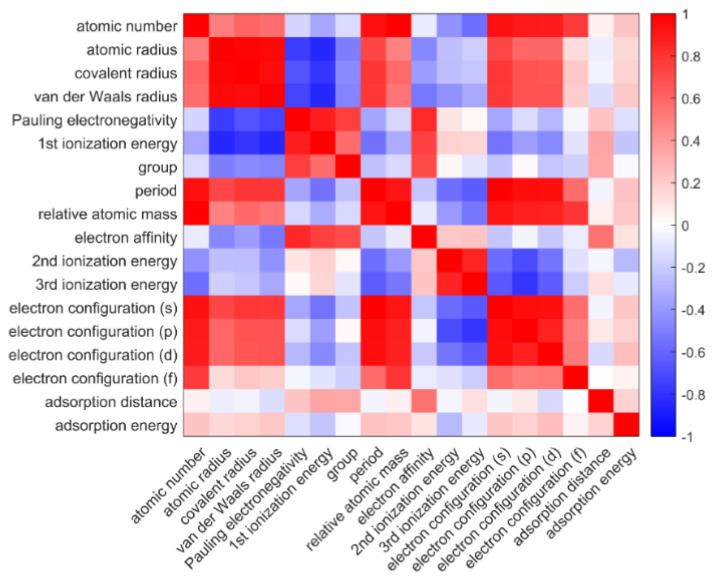
The heat map of Pearson correlation coefficient matrix among both the input features and the first principle calculated adsorption performances for adatom on graphene. The colors (blue and red) represent the positive and negative correlations.

**Figure 3 materials-16-02633-f003:**
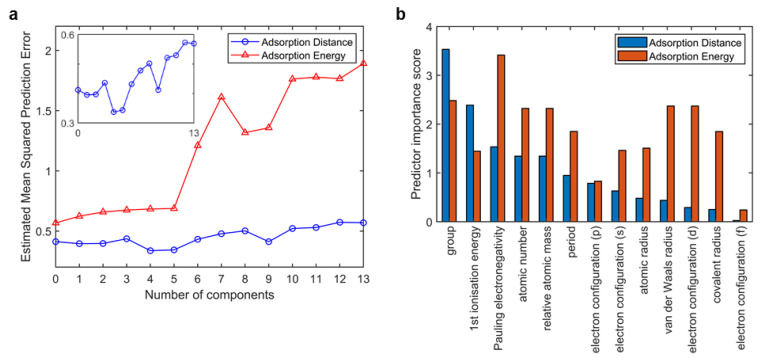
Feature selection. (**a**) Prediction error curves with number of inputs (the detailed plot of adsorption distance output is shown in northeast) with principal components regression (PCR) and (**b**) feature importance bar using F-tests. In both subfigures, the red bar represents adsorption energy output and the blue bar represents adsorption distance output.

**Figure 4 materials-16-02633-f004:**
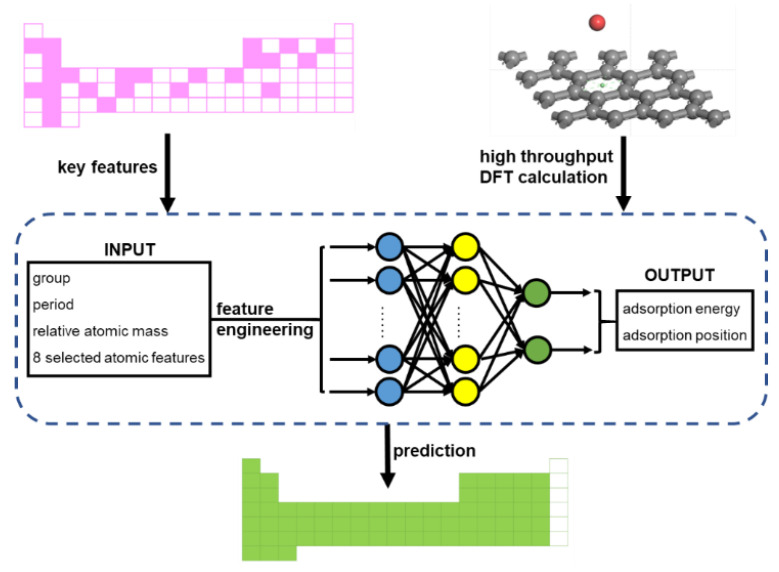
Scheme of the machine learner for the adatom adsorption problem. The workflow utilizes neural networks to accelerate the efficient first-principle calculation of adatom adsorption on graphene. Starting from the DFT calculation of random adatom adsorptions on graphene, the adsorption performances of all the adatoms on the periodic table (except noble gases, lanthanides, and actinides) were predicted using the network shown in the blue box.

**Figure 5 materials-16-02633-f005:**
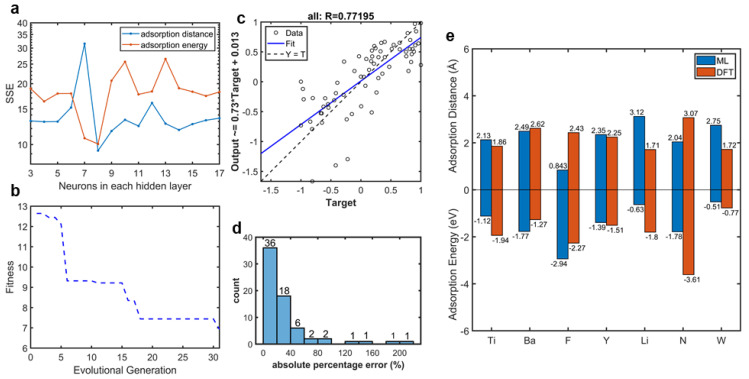
Network behaviors. (**a**) The sum of squared error (SSE) of the 5×n×2 network with different neurons in the hidden layer. (**b**) Fitness curve of our BPNN optimized by GA. (**c**) Regression scatter of the entire dataset (all the data normalized to −1 to 1), whose correlation coefficient between outputs (BPNN predicted adsorption properties) and targets (DFT-calculated adsorption properties) was 0.77195. The regression equation is shown on *y* axis. (**d**) Distribution of absolute percentage error of our BPNN with the whole dataset. (**e**) Comparation of adsorption properties (including adsorption energy and distance) of testing adatoms on graphene calculated by DFT (red bars) and our ML model (blue bars).

**Figure 6 materials-16-02633-f006:**
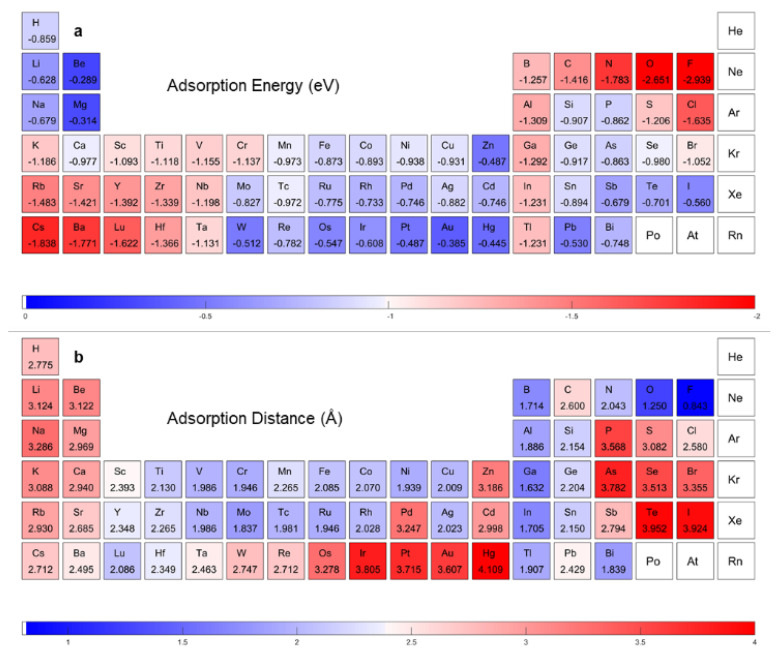
Adsorption properties prediction with our network. ML accelerated DFT-calculated (**a**) adsorption energy and (**b**) adsorption distance.

**Table 1 materials-16-02633-t001:** DFT-calculated interfacial distance ($d_{inter}$) and cohesive energy ($E_{coh}$) of graphene/Mg + M interfaces.

**Interface**	$d_{inter}^{Mg}$ (Å)	$d_{inter}^{M}$ (Å)	$E_{coh}$ (eV)	$E_{ads}^{ML}$ (eV)
Graphene/Mg+Zn	3.13	3.40	6.340	−0.487
Graphene/Mg+Li	3.04	3.42	6.377	−0.628
Graphene/Mg+Ca	3.27	2.82	6.379	−0.977
Graphene/Mg+Al	3.27	3.65	6.470	−1.309

## Data Availability

All data generated or analyzed in this study are included in this manuscript and Appendix A.

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
