# Peer review of "Accelerating Density Functional Calculation of Adatom Adsorption on Graphene via Machine Learning"

_materials, 2023, doi:10.3390/ma16072633_

Round 1

Reviewer 1 Report

The manuscript by Nan Qu et al. which is called “Accelerating density functional calculation of adatom adsorption on graphene via machine learning“ informs us about the DFT calculation of adatom adsorption on graphene. The motivation is written in the introduction. We believe that the references do not have to be so comprehensive, they can be reduced. Otherwise, the article is well-written without major errors and I recommend it for publication.

Author Response

Thanks for the suggestion and recommendation. We reduced the number of references from 45 to 32.

Reviewer 2 Report

The present manuscript reports on the “Accelerating density functional calculation of adatom adsorption on graphene via machine learning”. The work is of some interest. I recommend minor revision.

Following are some of the comments/suggestions which will be useful to the authors.

1. First of all, there are many previous works published for adsorption process. The authors seem deliberately avoid those papers. This is unusual, as the authors need to acknowledge the previous literature and compare their work with the others in the literature and demonstrate their research outcomes in terms of advantages and disadvantages. Some of studies are given below need to cited; doi.org/10.1016/j.jece.2023.109270; doi.org/10.1515/zpch-2022-0038; doi.org/10.1039/C5RA05785J; doi.org/10.1016/j.cplett.2020.137645.

2. The conditions during performance data is missing from each figure.

3. Why adsorptive energy increases very low in starting values, rapidly after that, decreases and again starts to increases as given in figure 3. The explanation of this behavior is insufficient.

4. Improve the title of the manuscript.

Reviewer 3 Report

The authors propose a workflow for quick adsorption structure selection using a combination of first principles calculation and machine learning. They demonstrate the effectiveness of the workflow using the example of single-atom adsorption on graphene and its application to graphene/Mg composites design. The workflow offers a quick structure search and properties prediction of adsorption problems across the periodic table, with only one third of the element cases, saving about two thirds computing cost.

Based on the thorough organization of the paper and the comprehensive analysis of the results, I strongly recommend that the paper be considered for publication in Materials. However, I kindly request that the authors address the following comments before finalizing the submission.

1.     What is the proposed workflow for adsorption structure selection?

2.     What kind of predictions can be made using this workflow and how accurate are they?

3.     Can the workflow be used for other adsorption problems?

4.     Can the authors provide charge transfer plots or Bader charge analysis to understand the adsorption process better?

5.     The mean percentage error of this model is below 35%. Can the authors suggest methods to improve its accuracy?

6.     Can the current model predict the adsorption and dissociation of molecules on surfaces?

Round 2

Reviewer 3 Report

The authors have modified the manuscript after the reviewers' suggestions. The manuscript can be accepted in the present form.